# The Effects of Exercise Duration and Intensity on Breast Cancer-Related DNA Methylation: A Randomized Controlled Trial

**DOI:** 10.3390/cancers13164128

**Published:** 2021-08-17

**Authors:** Arielle S. Gillman, Timothy Helmuth, Claire E. Koljack, Kent E. Hutchison, Wendy M. Kohrt, Angela D. Bryan

**Affiliations:** 1Center for Health and Neuroscience, Genes, and Environment (CUChange), Department of Psychology & Neuroscience, University of Colorado Boulder, Boulder, CO 80309, USA; Timothy.helmuth@colorado.edu (T.H.); kent.hutchison@colorado.edu (K.E.H.); angela.bryan@colorado.edu (A.D.B.); 2University of Colorado Anschutz Medical Campus, Aurora, CO 80045, USA; Claire.koljack@cuanschutz.edu (C.E.K.); wendy.kohrt@cuanschutz.edu (W.M.K.)

**Keywords:** breast cancer, exercise, physical activity, methylation

## Abstract

**Simple Summary:**

While physical activity has been associated with reduced cancer risk, it is not well understood why this is the case. One possible reason is that physical activity affects *DNA methylation*—a process that functions to turn certain genes “on” or “off”—which can affect cancer-related processes in the body. We tested this in an experimental study, where women aged 30–45 were randomly assigned to complete 16 weeks of exercise of varying intensity and duration. We hypothesized that higher levels of exercise would lead to changes in DNA methylation that would be associated with reduced cancer risk. Contrary to our hypotheses, we found that the total amount of exercise completed was not associated with changes in DNA methylation, though we did find that increases in VO_2_max, a marker of physical fitness, were associated with decreases in methylation of the *BRCA1* gene, and higher levels of exercise during a follow-up period were associated with lower levels of methylation of the *GALNT9* gene. This study provides preliminary evidence that increased exercise behavior or fitness may affect the methylation of some genes that are related to breast cancer.

**Abstract:**

Emerging research suggests that one mechanism through which physical activity may decrease cancer risk is through its influence on the methylation of genes associated with cancer. The purpose of the current study was to prospectively test, using a rigorous experimental design, whether aerobic exercise affects DNA methylation in genes associated with breast cancer, as well as whether quantity of exercise completed affects change in DNA methylation in a dose–response manner. 276 women (M age = 37.25, SD = 4.64) were recruited from the Denver metro area for a randomized controlled trial in which participants were assigned to a supervised aerobic exercise program varying in a fully crossed design by intensity (55–65% versus 75–85% of VO_2_max) and duration (40 versus 20 min per session). DNA methylation was assessed via blood samples provided at baseline, after completing a 16-week supervised exercise intervention, and six months after the intervention. 137 participants completed the intervention, and 81 had viable pre-post methylation data. Contrary to our hypotheses, total exercise volume completed in kcal/kg/week was not associated with methylation from baseline to post-intervention for any of the genes of interest. An increase in VO_2_max over the course of the intervention, however, was associated with decreased post-intervention methylation of *BRCA1*, *p* = 0.01. Higher levels of self-reported exercise during the follow-up period were associated with lower levels of *GALNT9* methylation at the six-month follow-up. This study provides hypothesis-generating evidence that increased exercise behavior and or increased fitness might affect methylation of some genes associated with breast cancer to reduce risk.

## 1. Introduction

In 2020, an estimated 276,480 women will be diagnosed with breast cancer, and 42,170 will die from breast cancer [1]. Breast cancer is the most common cancer diagnosis in the United States, representing 15.3% of all new cancer diagnoses [1]. Effective prevention strategies thus have the potential to have enormous effects on morbidity and mortality due to breast cancer.

Physical activity has been associated with a reduced risk of developing many cancers including cancer of the breast [2,3,4,5,6], but the exact biological mechanisms are not completely understood [7,8]. One promising hypothesis is that physical activity may decrease cancer risk via its influence on the methylation of genes associated with cancer [9,10]. DNA methylation is an epigenetic process wherein a methyl group is added to the 5′ position of the cytosine pyrimidine ring within cytosine-phosphate-guanine (CpG) dinucleotides. Along with other epigenetic mechanisms, such as histone modifications, DNA methylation essentially functions as a “switch” that turns certain genes on and off, a mechanism that is crucial for development, differentiation, and genomic stability. Many functional genes have very high concentrations of CpGs in their promoter regulatory region, the region that controls gene expression. In most normal cells, these CpG “islands” are unmethylated or have low levels of methylation. In cancer, this healthy state is disrupted, such that higher levels of methylation may silence the action of tumor suppressor genes that prevent the proliferation of cells that characterizes tumor development [11,12,13,14,15,16]. There is also evidence of genes for which hypomethylation is associated with cancer. Oncogenes, which are normally silenced, are activated when they become less methylated [17,18]. Finally, hypermethylation of Toll-like receptor genes (e.g., *TLR4* and *TLR6*) associated with inflammation (a cancer-related process [19,20,21]) and breast cancer cell survival/proliferation can reduce their expression, thus reducing the inflammatory signaling associated with chronic illness and tumor cell survival [22,23].

Emerging evidence suggests that it has become possible to detect methylation patterns that can predict the development of certain cancers among at-risk individuals before clinical diagnosis (e.g., [24]). Thus, enhancing our understanding of DNA methylation as it relates to cancer and the processes by which methylation may be influenced is of great value.

Body mass index (BMI) has been associated with methylation of candidate genes for cancer [25], while elevated weight status and poorer aerobic fitness were associated with hypomethylation of inflammatory genes [26]. Physical activity levels [27] and cardiorespiratory fitness assessed by maximal aerobic power (VO_2_ max) [28] are associated with differential methylation of genes specific to breast cancer. Physical activity has been shown to be inversely associated with promoter hypermethylation of the APC tumor suppressor gene in women without breast cancer [27]. One study found a positive association between physical activity and global genomic DNA methylation, but it was attenuated when adjusted for covariates [29], and another found no differences in DNA methylation of candidate genes related to breast cancer between the aerobic exercise and control arms of a physical activity intervention in postmenopausal women [30]. Thus the nature and extent of the effect of physical activity on methylation of genes associated with breast cancer risk remains an open question, and the optimal amount of physical activity for producing these effects, in terms of intensity, duration, and frequency, also remains unclear.

The purpose of this study was to prospectively test, using a rigorous experimental design, whether aerobic exercise influences DNA methylation in genes associated with breast cancer, as well as whether a dose response relationship exists between quantity of exercise completed and the degree of change in DNA methylation. To that end, we randomly assigned sedentary but otherwise healthy pre-menopausal women to a sixteen- week supervised exercise intervention that varied by intensity and duration. We collected blood at baseline and post-intervention to assess changes in DNA methylation. We collected DNA again six months after the exercise training, in order to explore the potential durability of changes in methylation.

The first aim was to examine changes in DNA methylation from baseline to post-intervention. We hypothesized a dose–response relationship such that women who completed the highest quantity of exercise would experience the greatest improvements in methylation. The second aim of the study was to examine the potential that the positive effects of exercise on methylation might revert back to previous levels if exercise was not continued. Due to high levels of participant attrition and limited viable samples for methylation data, the sample size for pre-post methylation was lower than expected (detailed below). Thus, we consider the analyses presented in the current paper to be exploratory and hypothesis-generating, with the goal of providing foundational knowledge for future studies that may test exercise as a way to facilitate breast cancer prevention.

We assessed methylation for eleven candidate genes selected for their association with breast cancer outcomes or inflammatory and immune responses associated with cancer, and/or because preliminary research showed associations between physical activity and/or VO_2_max and methylation at CpG cites for these genes [28]. These genes included *BRCA1*, a tumor suppressor gene that normally functions to repair double-stranded DNA breaks [18]; *RUNX3*, a tumor suppressor that has been shown to be inactivated in the early stages of breast cancer [31]; *PAX6*, a transcription factor/tumor suppressor which has been shown to be highly expressed in breast cancer cell lines [32]; *GALNT9*, which has been implicated in uncontrolled proliferation, invasion, and metastasis [33]; *SIM1*, a potential tumor suppressor found to be down-regulated in breast cancer and obesity [34]; *FBLN2*, an extracellular matrix protein found to be downregulated in breast cancer [35]; *AURKA*, an oncogene that propagates cell division and is up-regulated in cancer tissue [36]; *BCAR1*, which regulates cell growth and migration and is related to antiestrogen and chemotherapeutic resistance [37]; *BPIF4AP/BASE*, found to be more expressed in breast cancer [38]; and inflammation Toll-like receptor genes *TLR4* and *TLR6*, implicated in pathogen recognition and activation of the innate immune system and linked to breast cancer cell proliferation [22,23].

For the tumor suppressor genes *BRCA1*, *RUNX3*, *PAX6*, *GALNT9*, *SIM1*, and *FBLN2* [26,35,39,40,41,42], decreased methylation as a result of exercise was hypothesized; for the oncogenes *AURKA*, *BCAR1*, and *BPIF4AP/BASE* [37,38,43], increased methylation as a result of exercise was hypothesized; and for the inflammation Toll-like receptor genes *TLR4* and *TLR6*, increased methylation as a result of exercise was hypothesized [23,26].

## 2. Materials and Methods

### 2.1. Design

Participants were randomly assigned to one of four supervised aerobic exercise programs that varied in a factorial design by intensity (55–65% versus 75–85% of VO_2_max) and duration (40 versus 20 min per session) and allocated evenly across conditions. A laboratory-based graded maximal aerobic power treadmill test (VO_2_max) was conducted and participants provided blood samples for assessment of DNA methylation at baseline. Participants were then asked to come to our exercise facility four times per week for sixteen weeks to engage in supervised exercise at their randomly assigned intensity and duration. Participants repeated baseline measures at the 16-week laboratory visit, after completing the supervised exercise intervention. Six months after the end of the exercise intervention, participants provided a final blood sample and completed self-reports of recent exercise. This randomized controlled trial was registered at ClinicalTrials.gov: registration number NCT02032628.

### 2.2. Participants

276 women (Mean age = 37.25, SD = 4.64) were recruited from the Denver metro area at baseline. Inclusion and exclusion criteria are described in Table 1. Of the 276 participants initially enrolled, 219 began exercise sessions, and 135 participants finished the 16 weeks of exercise sessions and post-exercise measures. We observed some systematic differences between participants who completed the study compared to those who enrolled in the study but did not complete it. Specifically, participants who did not complete the study had lower levels of education, on average, and lower VO_2_max scores at baseline (see Table 2). Of the participants who completed the study, 81 had viable post-exercise methylation data and are included in the analyses for the current paper, and 88 participants had six-month follow-up methylation data. Methylation data were incomplete due to a combination of missing blood samples, issues the research team faced with transporting the blood between research facilities that led to difficulties extracting DNA from blood samples, and insufficient methylation signal during pyrosequencing. For CONSORT diagram, see Figure 1. Recruitment began on 1/30/2014 and six-month follow-up procedures were complete as of 8/17/2017. Descriptive statistics for the analytic sample are included in Table 3.

### 2.3. Measures

#### 2.3.1. Demographics

Data were assessed via an online survey administered via REDCap [44].

#### 2.3.2. VO_2_max

During an initial five-minute warm-up, the participant began walking at 2 mph on the treadmill and speed was adjusted to elicit a heart rate that was ~70% of their age-predicted maximal heart rate. Speed was then held constant and treadmill grade was increased by 2% every 2 min until volitional exhaustion. VO_2_max, measured in milliliters of oxygen consumption per kilogram of body weight per minute (mL/kg/min), was assessed using online computer-assisted open-circuit spirometry (ParvoMedics, Sandy, UT, USA) and confirmed by a respiratory quotient ≥1.1 and/or a detected plateau in VO_2_. VO_2_max was measured at baseline and after the intervention.

### 2.4. Exercise Volume Calculation

As is common in exercise studies, perfect compliance between the exercise prescription and actual training completed was not observed. While participants were expected to come in four times per week for 16 weeks (totaling 64 sessions), the actual number of sessions completed ranged from 25 to 64 (*M* for analytic sample = 52.02). Therefore, using data on attendance, condition assignment, and VO_2_max, we computed a total exercise volume score for each participant in kcal/kg/week using ACSM Guidelines [45].

### 2.5. BMI

Height and weight were measured to calculate body mass index (BMI) in kg/m^2^.

### 2.6. Gene Methylation

Methylation of CpG sites for the selected genes was assessed via pyrosequencing performed at EpigenDx (Worcester, MA, USA) using previously published procedures [26,46]. Regions and locations of CpG sites interrogated relative to respective transcriptional start site for each gene are described in Table 4. Data were presented as percent methylation at each of the CpGs. Similar to prior work [28], and to reduce alpha inflation due to the number of tests conducted, the percent methylation at each CpG was averaged for each gene.

### 2.7. Follow-Up Exercise Behavior

Six months after the intervention, self-reported moderate intensity exercise behavior in minutes was measured using the Stanford 7-Day Physical Activity Recall (PAR [47]).

### 2.8. Procedures

The study was approved by the Colorado Multiple Institutional Review Board and the Colorado Clinical & Translational Sciences Institute’s Scientific Advisory and Review Committee and all study procedures were conducted in accordance with universal ethical principles. Participants were recruited through advertisements in local publications, flyers posted in community locations, and digital advertisements in online forums (e.g., Craigslist, Facebook). Advertisements described the opportunity to participate in a program designed to help women begin an exercise program.

Interested participants contacted the research team to learn the study details and complete the eligibility assessment. Screened participants were scheduled for their first appointment where they were enrolled in the study by a professional research assistant, gave informed consent, and completed medical screening by a study physician. Eligible participants then completed both a self-report of physical activity over the past six months as well as the PAR to verify that they met exercise criteria. If eligible, they completed baseline assessments online via REDCap [44]. Finally, two venous blood samples (2.5 milliliters/sample) were drawn by a study nurse. One blood sample was transferred to a BD Vacutainer^®^ CPT Mononuclear Cell Preparation Tube (BD Diagnostics, Franklin Lakes, NJ, USA). The second blood sample was transferred to a PAXgene Blood RNA tube (Qiagen, Valencia, CA, USA) for future transcriptome analyses. Within 2 h of collection, blood samples from BD Vacutainer^®^ CPT Mononuclear Cell Preparation Tubes were centrifuged and processed according to the manufacturer’s protocol. Purified PBMC suspensions were stored at −70 °C. Frozen PBMCs and saliva samples were transferred to our genetics laboratory for DNA extraction and processing. DNA was extracted per the manufacturer’s instructions using DNA Genotek’s prepIt DNA extraction kit (Cat. Nos. PT-L2P-5 or PT-L2P-45). The DNA was quantified using Invitrogen’s Quant-iT™ PicoGreen dsDNA Kit (Cat. No. Q-33130) and cryogenically stored at −80 °C. Participants returned for another study visit where they completed the VO_2_max test.

The principal investigator/statistician used an online random number generator to generate the random allocation sequence. When a participant arrived for her first exercise session after the VO_2_max test, a professional research assistant assigned her to the next condition in the randomly generated list. Based on the VO_2_max test, the parameters of the exercise prescription (i.e., the intensity corresponding to the % of their own VO_2_max based on condition assignment) were set by the study exercise physiology team.

For each session, participants came to the facility and engaged in cardiovascular exercise (treadmill walking/running and occasionally elliptical trainers) at the prescribed intensity and duration. Participants wore heart rate monitors during each exercise session to verify prescribed intensity. At the end of the 16-week exercise intervention, participants completed the second VO_2_max graded treadmill test and blood draw. Six months after completion of the exercise intervention, participants completed the PAR and had blood drawn for methylation analyses.

### 2.9. Statistical Analysis

The sample size was selected to permit analysis of the intensity by duration interaction effect on changes in methylation at a two-tailed alpha of 0.05 and power level of at least 0.80 following Cohen [48] using G*Power 3.0.372. Assuming a moderate effect size for the interaction based on previous studies [28,49], 45 participants per group for a total N of 180 were required. To be fully powered at four months, accounting for dropout at all stages, we planned to recruit an initial sample of 300 women. Ultimately, 276 women were recruited due to time and resource availability. Given that our final sample size was notably lower than the a priori power analysis indicated was required, we consider between-group analyses conducted with the available data to be exploratory and hypothesis-generating.

Descriptive linear and quadratic patterns of methylation change were assessed using a repeated measures analysis of covariance (ANCOVA) where time of methylation assessment (pre-intervention, post-intervention, six months) is a repeated measures variable and age and BMI (mean-centered) are included as covariates [26,50]. To assess the dose–response effect of exercise quantity on change in methylation during the intervention, regression models for each of the eleven genes of interest were estimated. Post-intervention methylation was regressed on baseline methylation and the quantity of exercise completed during the intervention, again covarying age and BMI. Quantity of exercise completed during the intervention was operationalized in three ways: exercise volume in kcal/kg/week, change in VO_2_max, and duration/intensity condition assignment, resulting in three regression models for each gene. To explore methylation change after the intervention, regression models were estimated with six-month methylation for each gene of interest regressed on post-intervention methylation, covarying self-reported exercise at the six-month follow-up, age, and BMI.

## 3. Results

### 3.1. Change in Methylation: Baseline, Post-Intervention, and Follow-Up

Figure 2 depicts the change in methylation from baseline to post-intervention to follow-up across condition. For *AURKA* methylation, there was a significant linear effect such that methylation increased over time, *b* = 0.33, 95% C.I. [0.02, 0.64], *p* = 0.04, and no significant quadratic effect. *BCAR1* methylation also increased linearly over time, *b* = 0.36, 95% C.I. [0.10, 0.62], *p* = 0.006, with no quadratic effect. These results suggest that exercise is associated with healthy changes in methylation for *AURKA* and *BCAR1* that persist after the intervention formally ended. For *BPIF4AP* methylation, there was no linear change over time, *p* = 0.89, but a significant quadratic effect, *b* = 0.41, 95% C.I. [0.13, 0.69], *p* = 0.005, such that methylation increased from baseline to post-intervention (a healthy change during exercise) and then decreased at follow-up (an unhealthy change after the intervention ended). *BRCA1* methylation increased linearly over time, *b* = 0.06, *p* = 0.04, 95% C.I. [0.002, 0.11] with no quadratic effect of time, *p* = 0.26, which suggests worsening *BRCA1* methylation overall during the study. *SIM1* methylation increased linearly over time, *b* = 0.23, 95% C.I. [0.11, 0.34], *p* < 0.001, and there was a significant quadratic effect, *b* = −0.11, 95% C.I. [−0.19, −0.03], *p* = 0.004, such that methylation did not change from baseline to post-intervention but increased (an unhealthy change) from post-intervention to follow-up. There were no overall significant linear or quadratic changes in methylation over time for *FBLN2*, *GALNT9*, *PAX6*, *RUNX3*, *TLR4*, or *TLR6*.

### 3.2. Change in Methylation from Baseline to Post-Intervention Based on Quantity of Exercise Completed

#### 3.2.1. Exercise Volume

Exercise volume calculated in kcal/kg/week was not associated with post-intervention methylation for any of the candidate genes.

#### 3.2.2. VO_2_max

*BRCA1* methylation was significantly negatively associated with change in VO_2_max, *b* = −0.05, 95% C.I. [−0.10, −0.01], *t*(66) = −2.52, *p* = 0.01. That is, individuals who increased their cardiovascular fitness displayed *less* of an increase in *BRCA1* methylation over time (see Table 5). We found a similar, though not significant, trend in the same direction for the methylation of tumor suppressor gene *FBLN2*, *b* = −0.08, 95% C.I. [−0.117, 0.01], *t*(67) = −1.80, *p* = 0.08. There were no other significant relationships between VO_2_max and change in methylation for the other genes of interest.

#### 3.2.3. Exercise Condition

There were no significant effects of condition on change in methylation.

### 3.3. Six-Month Follow-Up Methylation Change as a Function of Level of Continued Exercise

Participants reported an average of 133 weekly minutes of exercise via the PAR at follow-up (*SD* = 161.84, Range = 0–720). We regressed six-month methylation levels on self-reported exercise at six months, controlling for post-test methylation for that same gene, BMI, and age. Higher levels of self-reported exercise during the follow-up period were associated with lower levels of methylation of the tumor suppressor gene *GALNT9*, *b* = −0.006, 95% C.I. [−0.01, −0.001], *t*(48) = −2.42, *p* = 0.02 at the six-month follow-up. No other effects were evident.

## 4. Discussion

This study sought to explore whether higher volume (duration + intensity) of exercise was associated with healthy changes in methylation for genes previously identified to be associated with breast cancer. The results were mixed.

For some genes (*AURKA*, *BCAR1*), the data suggest modest positive changes, or at least the prevention of negative changes (*BPIF4AP*, *SIM1*), during an exercise intervention. For others the changes were moderated to some degree by dose of exercise. Contrary to hypotheses, we did not observe significant relationships between total exercise volume completed or condition assignment and methylation for any of the genes of interest. However, we observed a significant relationship between change in VO_2_max over time (which was significantly associated with total volume completed, *p* = 0.02) and methylation of the *BRCA1* gene in the expected direction. While we did not observe any other significant effects of change in VO_2_max on methylation of other genes, it is notable that the gene for which we did observe a signal is one that is arguably most well-known for its association with breast cancer risk [51,52,53,54,55].

We also found that participants who reported more minutes of exercise during the six-month follow-up had lower levels of *GALNT9* methylation at follow-up, which is a healthy pattern of methylation change for this tumor suppressor gene. This finding provides preliminary evidence that maintaining exercise behavior may have long-term effects on methylation, at least for this particular gene.

Overall, these analyses provide some support for the idea that DNA methylation may be one mechanism by which exercise behavior may reduce the risk of breast cancer. Importantly, the fact that we only observed effects of exercise on two of the candidate genes of interest provides support for the idea that DNA methylation is not a monolith—that is, changes in the exact same behavior may have different effects even on genes that are known to have similar biological effects. These findings may call into question the practice of examining “global methylation” as an indicator of health/disease or as an effect of some external factor (e.g., exercise, diet, smoking, toxin exposure), which often demonstrates null results [56]. For example, Jabłońska and Reszka found that selenium exposure was unrelated to global DNA methylation but was related to the specific methylation of tumor suppressor genes [57]. Similarly, a review of epigenetics in the obesity domain showed no consistent association with global methylation but did find associations at specific sites [58]. Future work would benefit from the development of function-specific panels of epigenetic markers that, ideally, respond similarly to a particular external stimulus. Further, it remains an open question in the field as to whether our blood-derived methylation results can be transferred to other cell types relevant to cancer development. A recent paper by Zhang et al. suggests that blood-derived DNA methylation markers could mirror changes in difficult-to-access tissues relevant to cancer development [59]. While we were not able to study this particular question given our study methods, our results provide support for investigating these questions more directly in future studies.

This study had multiple strengths. First, prior studies linking exercise and methylation have been almost entirely retrospective, such that participants are asked to report about their exercise behavior over some extended retrospective period (in some cases even a lifetime [60]). A strong innovation of the current work was the use of a rigorous prospective experimental design with random assignment to condition that allows for causal statements regarding the influence of the level and duration of increases in aerobic exercise and cardiovascular fitness on consequent improvements in DNA methylation. We were able to find only one similar randomized experiment in the literature [49], and though the findings provided support for a causal role of exercise on methylation of the ASC gene (responsible for IL-1β and IL-18 secretion), the participants were older (mean age ~65), methylation data were only obtained post-exercise intervention so change could not be assessed, and the genes explored were not related to breast cancer. By investigating a different age group, a broader set of genes and CpG sites, and assessing pre-intervention levels of methylation so that change could be measured, our study adds to the knowledge base regarding the influence of exercise on methylation.

Another strength of the current study was that exercise behavior was objectively measured and directly observed. The exclusive reliance on self-reporting in previous studies is a serious limitation [61]. Further, changes in fitness (VO_2_max) were assessed objectively, both to validate doses of exercise and to determine which of these variables were most closely associated with DNA methylation.

At the same time, there were important limitations to the current study. As we have noted throughout the manuscript, our sample size of viable post-exercise and six-month follow-up methylation data was small relative to our original sample size, both due to participant attrition and issues extracting DNA from the samples we did have. It is likely that we were ultimately underpowered to detect significant effects. Thus, we consider these results to be preliminary and hypothesis-generating. We also observed some significant differences (in education and baseline VO_2_max) between participants who completed the study and those who enrolled but did not complete the study, which may limit the generalizability of our findings. Further, our methylation data was extracted from a mixture of leukocyte cell types. Since methylation patterns can vary between cell types, it is important to take into consideration the heterogeneity of our blood samples. In addition, since individual regressions were examined for each gene of interest, there is an important multiple testing issue. The current analyses, since we consider them exploratory, were not corrected for multiple testing. Finally, we did not have a no-exercise control group, thus, the changes we observed in methylation that were not modified by exercise condition may reflect changes due to increased exercise behavior or simply to time passing. It is also a possibility that our participants may have engaged in some additional exercise outside of the prescribed supervised exercise sessions, which may have affected the quality of our comparisons. Our sample included women between the ages of 30 and 45; thus, findings may not generalize to men or other age groups.

## 5. Conclusions

Despite its limitations, this study is significant and has important potential implications for breast cancer prevention. These data provide preliminary information regarding possible biological mechanisms by which higher levels of exercise are translated into a lower incidence of breast cancer. This study also lays important groundwork for a range of investigations into biomarkers of cancer risk and the development and assessment of interventions to influence those biomarkers in beneficial directions. While the current study does not provide a clear answer as to exactly what level of exercise volume produces changes in DNA methylation, which might be associated with a reduced risk of breast cancer, it provides important initial evidence for this research question, and highlights the importance of future work in this area.

## Figures and Tables

**Figure 1 cancers-13-04128-f001:**
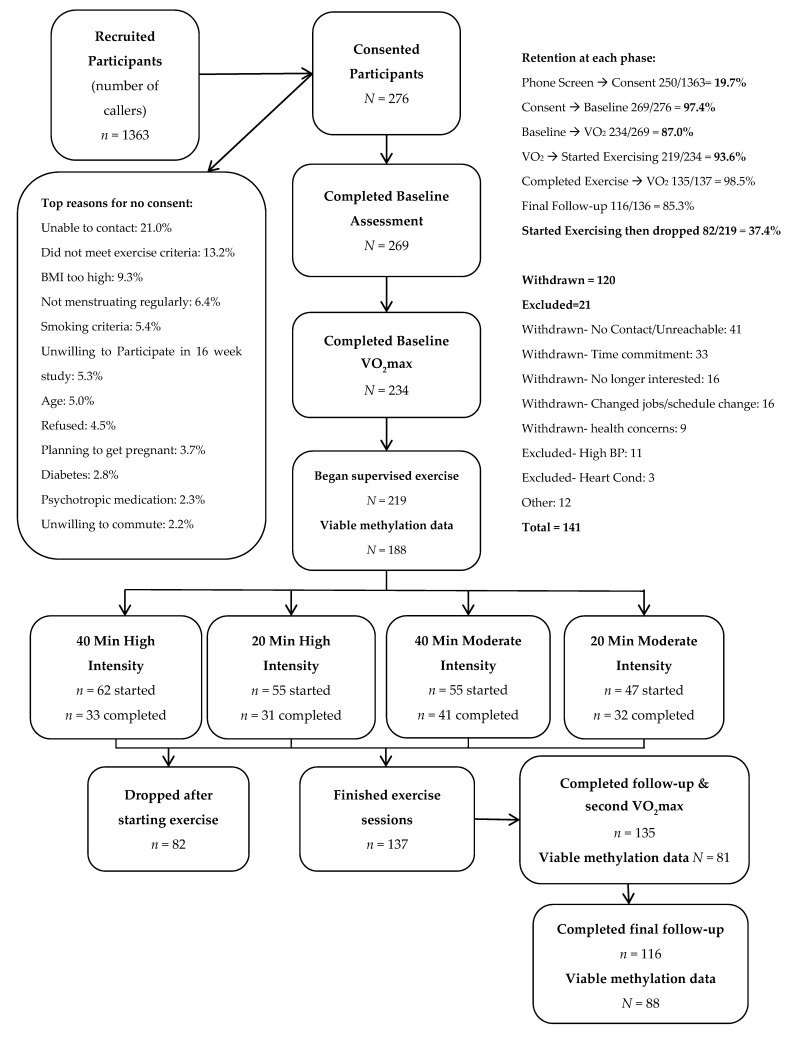
CONSORT diagram of participant flow throughout study.

**Figure 2 cancers-13-04128-f002:**
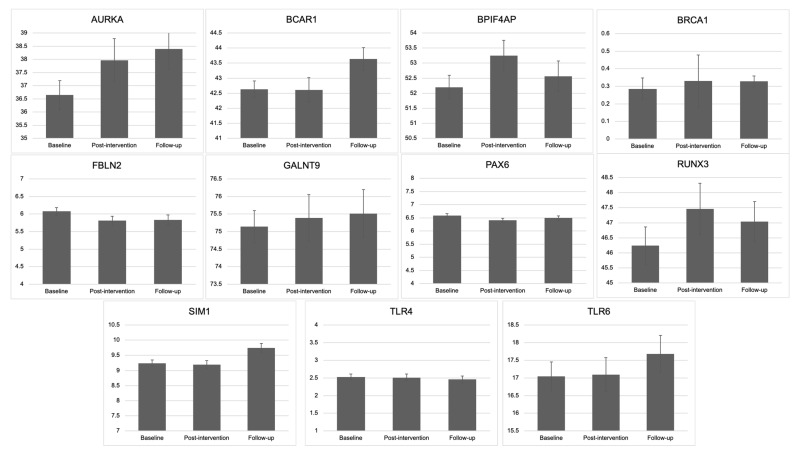
Average percent methylation for each gene at each time point, on average across condition. *Y*-axis for each panel represents % methylation. Bars are standard error bars.

**Table 1 cancers-13-04128-t001:** Study eligibility criteria.

Inclusion Criteria	Exclusion Criteria
Between ages of 30–45 <60 min of moderate intensity exercise per week Pre-menopausal Non-smoker Willing to accept random assignment Willing to provide blood samples Willing and physically capable of engaging in moderate exercise activity (i.e., no injuries, physical impairments, or pre-existing contraindications) as assessed by a study physician Ability to successfully complete a VO_2_Max test without evidence of cardiac or other abnormalities Planning to remain in the Denver metro area for the next 10 months	BMI > 39 kg/m^2^ Diabetic or on a restricted diet Controlled hypertension (resting systolic BP > 150 mmHg or diastolic BP > 90 mmHg) Cardiovascular or respiratory disease Serious arrythmias at rest during the VO_2_Max test Reported history of breast neoplasia Currently receiving treatment for any type of cancer Currently taking psychotropic medications except for depression and anxiety Currently under treatment for any psychiatric disorder Currently under treatment for alcohol or drug abuse Currently pregnant or attempting to become pregnant

Due to difficulty recruiting eligible participants, two changes to inclusion/exclusion were made early in the trial. The original cutoff for current exercise was 45 min or less and it was increased to 60. Original inclusion criteria stated women must have a “regular menstrual cycle” and this unintentionally excluded women on birth control that affects cycling. Thus, inclusion criteria were amended to include women using birth control.

**Table 2 cancers-13-04128-t002:** Demographic and participant characteristics for study completers vs. non-completers.

Characteristic	Completed Trial (*n* = 135) M (SD)	Enrolled, Not Completed (*n* = 141) M (SD)	Test Statistic for Group Differences
Age	37.43 (4.71)	37.08 (4.59)	*t*(267) = 0.60), *p* = 0.545
Race (% white)	59.7%	51.5%	*χ**^2^*(6) = 6.69, *p* = 0.035
Education (% college degree or higher)	71.6%	54.1%	*χ**^2^*(7) = 14.13, *p* = 0.05
BMI (kg/m^2^)	28.54 (5.62)	29.38 (5.15)	*t*(261) = −1.25, *p* = 0.211
BaselineVO_2_max (mL/kg/min)	27.95 (5.80)	26.46 (4.80)	*t*(229) = 2.08, *p* = 0.04
Self-reported exercise minutes/week	17.13 (32.61)	15.56 (25.02)	*t*(266) = 0.441, *p* = 0.66

M: Mean.

**Table 3 cancers-13-04128-t003:** Participant baseline characteristics.

Characteristic	Overall Sample (*N* = 81)	Low Intensity + 20 min (*n* = 21)	Low Intensity + 40 min (*n* = 23)	High Intensity + 20 min (*n* = 17)	High Intensity + 40 min (*n* = 20)
Age	37.14 (4.71)	37.00 (4.86)	37.26 (4.60)	37.82 (5.45)	33.55 (4.29)
Race (% White)	65.4%	61.9%	65.2%	70.6%	65.0%
BMI (kg/m^2^)	29.60 (5.69)	28.22 (6.02)	29.36 (4.95)	29.59 (5.69)	31.32 (6.08)
VO_2_max (mL/kg/min)	27.37 (5.29)	28.18 (5.73)	26.78 (5.38)	27.70 (5.41)	26.89 (4.85)
Self-reported exercise mins/week	18.27 (31.94)	12.86 (15.13)	22.85 (48.14)	17.65 (22.23)	19.25 (29.92)

Table represents baseline characteristics for the 81 participants with viable posttest methylation data. Baseline characteristics were not significantly different across conditions (all *p*’s > 0.38).

**Table 4 cancers-13-04128-t004:** Regions and locations of CpG site interrogated relative to respective transcriptional start site.

Gene	Gene Location of Methylation Profile	Region of Methylation Profile from Transcriptional Start Site (Number of Base Pairs)
*BRCA1*	5′ Untranslated Region	−76 to +23
*RUNX3*	5′ Untranslated Region	−12 to +76
*SIM1*	5′ Untranslated Region	+3 to +37
*AURKA*	5′ Untranslated Region	−1276 to −1241
*BCAR1*	5′ Untranslated Region	−503 to −482
*BPIFA4P*	5′ Untranslated Region	−529
*GALNT9*	Intron 1	+41,806 to +41,854
*FBLN2*	Intron 1	+263 to +268
*PAX6*	Intron 4	+12,543 to +12,609
*TLR4*	5′ Untranslated Region	+27 to +51
*TLR6*	5′ Untranslated Region	−1291 to −1269

**Table 5 cancers-13-04128-t005:** *BRCA1* (tumor suppressor) methylation at post-intervention regressed compared to baseline methylation, age, BMI, and VO_2_max change.

Variable	Coefficient	Std. Error	*T*	*p*
(Intercept)	0.189	0.090	2.08	0.041 *
Baseline methylation	0.974	0.050	19.60	<0.001
Age (centered)	0.026	0.015	1.78	0.080
BMI (centered)	−0.008	0.012	−0.69	0.491
Change in VO_2_max	−0.054	0.021	−2.52	0.014 *

Unstandardized regression coefficients. * *p* < 0.05.

## Data Availability

The data presented in this study are available on request from the corresponding author.

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
