# Peer review of "The Effects of Exercise Duration and Intensity on Breast Cancer-Related DNA Methylation: A Randomized Controlled Trial"

_cancers, 2021, doi:10.3390/cancers13164128_

Round 1

Reviewer 1 Report

The manuscript describes the results of a randomized controlled trial which was conducted to evaluate the effects of exercise duration and intensity on DNA methylation in cancer-related genes. The study findings were largely null, although there were some indications of elevated methylation in two oncogenes and declined methylation in one tumor suppressor gene. 

Comments:

The study was well designed, and 276 subjects were enrolled, but only 219 started the trial and 135 completed the entire trial. A comparison of demographics and available characteristics between those enrolled and completed and not completed (135 vs. 141) will be helpful in assessing if there are any systematical reasons. 

Also, among the 135 participants who completed the trial, only 81 subjects had complete methylation data.  It is unclear what the reasons are that made the methylation data unavailable for 54 subjects, missing blood samples or disqualified assay results or both. 

The authors may want to discuss the possibilities of outside physical activities and/or exercise occurred during the course of the trial which may contaminate the quality of comparisons.  

Author Response

Reviewer 1

The manuscript describes the results of a randomized controlled trial which was conducted to evaluate the effects of exercise duration and intensity on DNA methylation in cancer-related genes. The study findings were largely null, although there were some indications of elevated methylation in two oncogenes and declined methylation in one tumor suppressor gene. 

Comments:

The study was well designed, and 276 subjects were enrolled, but only 219 started the trial and 135 completed the entire trial. A comparison of demographics and available characteristics between those enrolled and completed and not completed (135 vs. 141) will be helpful in assessing if there are any systematical reasons. 

We appreciate the positive comment regarding the design of the study. Per the reviewer’s suggestion, we now provide a table that describes differences in demographic and other characteristics between participants who completed the full study and those who enrolled but did not complete. We did find that participants who did not complete the study were significantly less educated and had a lower baseline VO2max on average than participants who completed the trial. We include this table in the revised manuscript (now Table 2) and mention these differences in the Methods and discuss it as a limitation: “We also observed some significant differences (in education and baseline VO2max) between participants who completed the study and those who enrolled but did not complete the study, which may limit the generalizability of our findings” (p. 11).

Characteristic

Completed trial (n=135)

M (SD)

Enrolled, not completed (n=141)

M (SD)

Test Statistic

Age

37.43 (4.71)

37.08 (4.59)

t(267)=0.60), p=.545

Race (% white)

59.7%

51.5%

χ2(6)=6.69, p=.035

Education (% college degree or higher)

71.6%

54.1%

χ2(7)=14.13, p=.05

BMI

28.54 (5.62)

29.38 (5.15)

t(261)=-1.25, p=.211

Baseline VO2max

27.95 (5.80)

26.46 (4.80)

t(229)=2.08, p=.04

Self-reported exercise minutes/week

17.13 (32.61)

15.56 (25.02)

t(266)=0.441, p=0.66

Also, among the 135 participants who completed the trial, only 81 subjects had complete methylation data.  It is unclear what the reasons are that made the methylation data unavailable for 54 subjects, missing blood samples or disqualified assay results or both. 

Methylation data were incomplete due to a combination of missing blood samples, issues the research team faced with transporting the blood between research facilities which led to difficulties extracting DNA from blood samples, and insufficient methylation signal during pyrosequencing. We have added this information to the revised manuscript (p. 4).

The authors may want to discuss the possibilities of outside physical activities and/or exercise occurred during the course of the trial which may contaminate the quality of comparisons. 

We appreciate this suggestion. It is possible that our participants completed outside activities during the course of the trial—though participants were encouraged to complete all their exercise during the intervention in the exercise lab. Still, we have added the possibility of outside exercise as a potential limitation (p. 12): “It is also a possibility that our participants may have engaged in some additional exercise outside of the prescribed supervised exercise sessions, which may have affected the quality of our comparisons.”

Reviewer 2 Report

The present study investigates the role of exercise with different duration and intensity on DNA methylation relevant in tumor development in PBMCs. Participants provided blood samples for assessment of DNA methylation at before, after completing the 16-week supervised exercise intervention, and six months after the intervention. The isolated PBMCs are analyzed for gene specific methylation pattern. It is shown that gene metyhlation of BRCA1 and GALNT9 are altered in relation to VO2max or to self-reported exercise during the follow-up period. While the trainings volume wasn’t correlated with DNA methylation changes. The correlation with VO2max gives evidence that the fitness level is an important factor for methylation of genes related to tumor development. The results are interesting and the study is performed state of the art, there are principal limitations given in the specifc comments.

Specific comments:

  1. Although the study is state of the art and the most studies use PBMCs. It is a heterogenous cell population which can be changed in dependence of intervention. Methylation pattern can vary between specific cell types. This aspect should be at least discussed.
  2. It should be discussed if the methylation results can be transfered to other cell types which are relevant in cancer development.
  3. It should be more discussed which role the investigated and altered genes play in different cell types, especially the cell which are in the PBMC-fraction.

The study is interesting and value to be published but the discussion shopuld be adapted and more cell type specific discussion is necessary as well as discuss about further limitations.

Author Response

Reviewer 2

The present study investigates the role of exercise with different duration and intensity on DNA methylation relevant in tumor development in PBMCs. Participants provided blood samples for assessment of DNA methylation at before, after completing the 16-week supervised exercise intervention, and six months after the intervention. The isolated PBMCs are analyzed for gene specific methylation pattern. It is shown that gene methylation of BRCA1 and GALNT9 are altered in relation to VO2max or to self-reported exercise during the follow-up period. While the trainings volume wasn’t correlated with DNA methylation changes. The correlation with VO2max gives evidence that the fitness level is an important factor for methylation of genes related to tumor development. The results are interesting and the study is performed state of the art, there are principal limitations given in the specific comments.

We thank the reviewer for the positive review of our study and are happy to address the outlined limitations.

Specific comments:

  1. Although the study is state of the art and the most studies use PBMCs. It is a heterogenous cell population which can be changed in dependence of intervention. Methylation pattern can vary between specific cell types. This aspect should be at least discussed.

We agree with the reviewer that this is a limitation of our study, since methylation patterns vary between specific cell types and our methylation data arises from a mixture of leukocyte cell types. That said, the purpose of our study was to examine the effects of exercise on methylation status in healthy individuals and to uncover possible broad mechanisms for which exercise may decrease cancer risk, i.e., rather than determining a particular biomarker for breast cancer or to discover in granularity how one particular cell type responds to an exercise intervention. Thus, we believe that although there are limitations to examining methylation patterns across multiple cell types, it is not a severe limitation in this context.

We have added this as a limitation in the Discussion section (p. 11): “Further, our methylation data was extracted from a mixture of leukocyte cell types. Since methylation patterns can vary between cell types, it is important to take into consideration the heterogeneity of our blood samples.”

  1. It should be discussed if the methylation results can be transferred to other cell types which are relevant in cancer development.

We appreciate this suggestion and have added some discussion of this topic to the manuscript (p. 11): “Further, it remains an open question in the field as to whether our blood-derived methylation results can be transferred to other cell types relevant in cancer development. A recent paper by Zhang et al. suggests that blood-derived DNA methylation markers could mirror changes in difficult-to-access tissues relevant to cancer development. While we were not able to study this particular question given our study methods, our results provide support for investigating these questions more directly in future studies.”

  1. It should be more discussed which role the investigated and altered genes play in different cell types, especially the cell which are in the PBMC-fraction.

We appreciate the reviewer’s comment. As we noted above, it is a limitation of our study that our methylation data was extracted from a mixture of leukocyte cell types. Thus, we examined methylation patterns, on average across cell types, but were not able to examine how methylation patterns differed between cell types in this study. Instead, our motivation was to examine methylation changes in the investigated genes as a proxy for broad mechanistic changes in methylation due to exercise, rather than necessarily how methylation changes in these genes may affect the PBMC fraction specifically.

Specifically, we selected the genes of interest based on previous work demonstrating a relationship between physical activity and methylation of these genes specifically (Bryan et al., 2012), which has been previously linked in the literature to breast cancer and/or inflammatory processes that relate to cancer (FBLN2: Hill et al., 2010; BRCA 1: Veeck & Esteller, 2010; RUNX3: Chen et al., 2012; SIM1: Kwak et al., 2007; AURKA: Du et al., 2021; BCAR1: Wallez et al., 2014; BPIFA4P or BASE: Bretschneider et al., 2008; GALNT9: Hussein et al., 2016; PAX6: Xia et al., 2015; TLR4/6: Akira et al., 2001; Yang et al., 2010).

In the Introduction of the revised manuscript (p. 3), we now describe in more detail the relationship between each gene of interest and breast cancer, and include additional citations: “These genes included BRCA1, a tumor suppressor gene that normally functions to repair double-stranded DNA breaks18, RUNX3, a tumor suppressor that has been shown to be inactivated in the early stages of breast cancer31, PAX6, a transcription factor/tumor suppressor which has been shown to be highly expressed in breast cancer cell lines32, GALNT9, which has been implicated in uncontrolled proliferation, invasion, and metastasis33, SIM1, a potential tumor suppressor found to be down-regulated in breast cancer and obesity34, FBLN2, an extracellular matrix protein found to be downregulated in breast cancer35, AURKA, an oncogene that propagates cell division and is up-regulated in cancer tissue36, BCAR1, which regulates cell grown and migration and is related to antiestrogen and chemotherapeutic resistance37, BPIF4AP/BASE, a found to be more expressed in breast cancer38, and inflammation toll-like receptor genes TLR4 and TLR6, implicated in pathogen recognition and activation of the innate immune system and linked to breast cancer cell proliferation22,23.”

The study is interesting and value to be published but the discussion should be adapted and more cell type specific discussion is necessary as well as discuss about further limitations.

We hope the changes we have outlined above have addressed the reviewer’s concerns, and we thank them for the suggestions which have improved our manuscript.
